# Fasting Protocols Do Not Improve Intestinal Architecture and Immune Parameters in C57BL/6 Male Mice Fed a High Fat Diet

**DOI:** 10.3390/medicines10020018

**Published:** 2023-02-17

**Authors:** Raed Y. Ageeli, Sunita Sharma, Melissa Puppa, Richard J. Bloomer, Randal K. Buddington, Marie van der Merwe

**Affiliations:** College of Health Sciences, The University of Memphis, Memphis, TN 38152, USA

**Keywords:** time-restricted feeding, alternate day fasting, high fat diet (HFD), plant based diet, mucosal immune system, intestinal morphology/histology

## Abstract

Background: The intestinal ecosystem, including epithelium, immune cells, and microbiota, are influenced by diet and timing of food consumption. The purpose of this study was to evaluate various dietary protocols after ad libitum high fat diet (HFD) consumption on intestinal morphology and mucosal immunity. Methods: C57BL/6 male mice were fed a 45% high fat diet (HFD) for 6 weeks and then randomized to the following protocols; (1) chow, (2) a purified high fiber diet known as the Daniel Fast (DF), HFD consumed (3) ad libitum or in a restricted manner; (4) caloric-restricted, (5) time-restricted (six hours of fasting in each 24 h), or (6) alternate-day fasting (24 h fasting every other day). Intestinal morphology and gut-associated immune parameters were investigated after 2 months on respective protocols. Results: Consuming a HFD resulted in shortening of the intestine and reduction in villi and crypt size. Fasting, while consuming the HFD, did not restore these parameters to the extent seen with the chow and DF diet. Goblet cell number and regulatory T cells had improved recovery with high fiber diets, not seen with the HFD irrespective of fasting. Conclusion: Nutritional content is a critical determinant of intestinal parameters associated with gut health.

## 1. Introduction

The increase in obesity and obesity-associated diseases has led to the implementation of various nutritional strategies to reduce or prevent weight gain and improve metabolic health. Many of these strategies include protocols that restrict food (e.g., vegetarian or low carbohydrate diet), calories, or time of food availability. Fasting protocols such as intermittent fasting have become very popular in recent years due to the benefits seen in metabolic health [1,2].

Time-restricted feeding (TRF) and intermittent fasting are common fasting protocols that mainly depend on restriction of feeding periods. The restriction can range from several hours every day as in TRF or a complete/partial fasting every other day which is referred to as alternate-day fasting (ADF) [2,3]. The effects of TRF and ADF on metabolic dysfunction and weight loss have been extensively studied in both animals and humans. In mice fed a high fat diet (HFD), TRF and ADF increased weight loss, decreased fat mass, and prevented adiposity, as well as improved glucose tolerance, insulin-and leptin resistance, and LDL cholesterol and triacylglycerol concentrations [2,4]. Similar results have also been observed in humans [3,5]. Restriction of calories (CR) is a dietary strategy that relies on reducing the number of calories consumed without taking fasting time into consideration [6]. CR is also associated cardiovascular disease, oxidative stress, and improved insulin resistance and lipoprotein profiles [2,7].

Plant-based dietary protocols have also shown efficacy for weight management [8]. Lower body mass index (BMI) and decreased calorie intake were observed in vegetarians vs. nonvegetarians [9]. Following a plant-based diet typically increases consumption of high-fiber/nutrient-dense whole foods instead of processed/animal-based foods [10]. This characteristic of the vegetarian diets is an important determinant of BMI and management of body weight [11,12]. Plant-based protocols can reduce low-density lipoprotein (LDL,) cholesterol, and L-carnitine metabolites linked to red meat consumption, as well as improve insulin resistance, body composition, and protect against type 2 diabetes [13,14].

In addition to the change in metabolic parameters, diet and fasting protocols also affect the intestinal microbial composition. While diet composition is the main driver of the microbiome assemblage, fasting is able to alter various microbial populations [15]. For example, the phylum Verrucomicrobia is increased, while Firmicutes is decreased, with fasting [16].

What is less well understood is the effect of diet and fasting on the intestinal morphology and associated immune cells. The intestinal epithelial layer is the initial site of contact between the host and food/microbes [17]. This single layer of cells is produced by stem cells located at the base of the intestinal crypts. Dividing stem cells migrate up towards the villi and differentiate into absorptive enterocytes, mucous secreting goblet cells, hormone releasing enteroendocrine cells, antibacterial Paneth cells, microfold cells, cup cells, and tuft cells [18]. The function of this layer of cells includes absorption of nutrients and mucus secretion to form a protective layer against microbes and antigens [19]. The type of diet and its components have been shown to alter the structural morphology of the intestine [20]. Studies in murine models demonstrated that caloric restriction while consuming a chow diet protects against intestinal shortening with aging [21], while high fat diets reduce overall length of small intestine and colon as compared to the normal-control and methionine-restricted diets [22]. Interestingly, the villi height and its ratio to crypt depth in the ileum are also decreased by HFD [22]. In the colon, high fat diets have been shown to cause crypt loss, ulceration, and goblet cells reduction in mice [23,24]. In contrast, high-fiber diets are able to reverse the impact that the HFD has on intestinal morphology [24,25].

The mucosal immunity of the intestine is also influenced by the diet and food-related antigens [26]. Forkhead box P3 (Foxp3) and retinoid orphan receptor (ROR)γt are transcription factors that influence polarization of T cells to either regulatory T cells (Tregs) or Th17 cells [27]. Tregs and Th17 cells are responsible for mediating the immune response through their suppressive and/or stimulative function [28]. Under the transcriptional regulation of RORγt, Th17 cells participate in the secretion of multiple inflammatory cytokines, such as Interleukin (IL)-17A, IL-17F, IL-21, and IL-22 [28], while Foxp3 is a transcriptional regulator of Tregs which in turn plays a role in the secretion of anti-inflammatory cytokines IL-10 and transforming growth factor (TGF)-β [28]. In animal models, the HF diet increased the expression of the proinflammatory cytokines in intestinal tissues, including tumor necrosis factor (TNF)-α, IL-1β, IL-22, and IL-6, as well as the circulating cytokines [29,30], while high fiber diet (inulin and oligofructose) increased IL-10 in Peyer’s patches and concentration of sIgA in cecum [31]. The cecal concentration of IL-1β was also decreased with consumption of inulin and oligofructose [32], while soluble dextrin fiber from tapioca and corn is able to decrease the secretion of the proinflammatory cytokines in an IL-10 deficient mice model [33].

As the role of various fasting protocols on the epithelial layer and mucosal immunity of the intestine remains unclear, we investigated the influence of various fasting protocols, while consuming a high fat diet, on intestinal morphology and immune parameters.

## 2. Materials and Methods

### 2.1. Experimental Animals and Diet

Sixty C57BL/6 male mice (four-week-old) were purchased (Envigo, Prattville, AL, USA) and housed at the USDA approved facility at the University of Memphis. Mice were treated as previously described [2,34]. All experiments were approved by the University of Memphis Institutional Animal Care and Use Committee (Protocol #0806). Animals were entrained to a reverse light–dark schedule (12-h dark, 12-h light) with lights off between 07:00 and 19:00 for 2 weeks with ad libitum access to standard rodent diet (Teklad global 2018; Envigo Laboratories Inc; 18% fat, 58% carbohydrates, 24% protein; 3.1 kcal/g).

An age- and sex-matched, healthy control group (CC, *n* = 8) consumed a standard rodent chow diet (Teklad Global 2018, Envigo Laboratories Inc.; 18% fat, 58% carbohydrates, 24% protein; 3.1 kcal/g) for the duration of the study. Fifty-two mice were first fed a high-fat diet (HF, D12451, Research Diets, Inc., New Brunswick, NJ, USA; 45% fat, 35% carbohydrate, and 20% as protein) for six weeks to induce weight gain and intestinal ecology change, and then randomly assigned to follow one of six different dietary protocols for 8 weeks; (1) ad libitum high-fat diet (HFD, *n* = 8), (2) standard rodent chow diet (SW-C, *n* = 9), (3) HF-calorie-restricted diet (CR, 20% reduction of total calories as compared to the HF group, *n* = 8), (4) HF-time-restricted feeding (TRF, 18:6 fasting protocols; the food access is during the first 6 h of active phase of mice, *n* = 9), (5) HF-alternate-day fasting (ADF, ad libitum access to food every other day, *n* = 9), and (6) plant-based diet high in fiber (including amylose, amylopectin and inulin) and omega 3 fatty acids resembling the human dietary strategy known as Daniel Fast (DF, *n* = 9) (Research Diets; product: D13092801; 59% carbohydrate, 15% protein, and 25% fat). Information about composition of both the HF and DF diet has previously been published [2,34]. Animals remained on their respective dietary protocols for 8 weeks. See Figure 1 for experimental set up. Three mice were lost during the study due to fighting and infection. At the end of 8 weeks, animals were sacrificed over a 3-day period. Animals were euthanized using isoflurane and cervical dislocation. The gastrointestinal tract was immediately removed, and length of small intestine (SI, stomach-cecum) and colon (cecum-rectum) measured. The cecum was removed and immediately weighed. After length of small and large intestine was measured, a section (approximately 1 cm) from the proximal, mid, and distal SI and colon were removed and stored in Formalin (Fisher Scientific Co. LLC, Dallas, TX, USA) until processed for histological staining. An additional section was also immediately frozen in liquid nitrogen and stored at −80 °C. The remaining colon were used for cell isolation. 

### 2.2. Tissue Sampling and Staining

After fixation, all tissues were dehydrated gradually in ethanol (1× 15 min in 70% and 90%; 2× 15 min in 100%; 1× 30 and 45 min in 100%). Subsequently, the tissues were cleared by Histoclear (HISTO-CLEAR II, # 64111-04, Electron Microscopy Sciences, Hatfield, PA, USA; 2× 20 min and 1× 45 min) and paraffin imbedded (Paraplast X-tra, #39603002, Leica Biosystems, Deer Park, IL, USA; 2× 30 min, and 1× 45 min at 60 °C). Samples were sectioned at 5-μm thickness. Slides were deparaffinized in 100% Histoclear (HISTO-CLEAR II, 2× 3 min), 1:1 Histoclear:ethanol (1× 3 min), and graded ethanol (1× 3 min). The tissues were than rehydrated in distilled water, and stained with hematoxylin (Hematoxylin+, =, Fisher Scientific, USA), and eosin (MilliporeSigma™ Eosin Y-Solution 0.5% Alcoholic, Fisher Scientific, USA) using a standard protocol [35]. Hematoxylin and eosin-stained slides were used for villi length and crypt depth determination. 

### 2.3. Periodic Acid-Schiff (PAS)-Alcian Blue (AB)

Alcian blue stain (Alfa AesarTM Alcian Blue 8GX, #AAJ6012209, Fisher Healthcare, Waltham, MA, USA) was used to identify goblet cells. The tissues were deparaffinized and dehydrated similar as in H&E staining, and then incubated in 3% glacial acetic acid solution for 3 min. Followed by 30-min incubation in an Alcian-blue stain solution (1% alcian blue in 3% glacial acetic acid; pH 2.5), and then washed. The slides were stained with Nuclear Fast Red Solution (Nuclear Fast Red Solution, # TS10-500, Tyr Scientific LLC, Wellsville, UT, USA) for 5 min and washed. 

### 2.4. Measurement of Villus Height, Crypt Depth, and Goblet Cells Count

For histological analysis, cross sectional images of the proximal, mid, and distal small intestine and colon were captured by an EVOS^TM^ 7000 Imaging System at a 10× magnification. Invitrogen Celleste Image Analysis Software (SKU# AMEP4816) was then used for villi and crypt morphometric evaluation. The lines feature was used to measure villi height from the top of villus to the top border of muscularis mucosae, while the crypt depth was measured form the top to the base of the crypt. Eleven to eighteen measurements were taken per mouse. An average for each mouse was then determined. The number of goblet cells was counted in both villi and crypt of distal small intestine and crypt of colon. The stained goblet cells were counted from three individual regions for each mouse and the average cell number per villi or crypt was then calculated.

### 2.5. Real-Time qRT-PCR

Intestinal tissue were homogenized in Trizol (TRIzolTM Reagent, # 15596018, Invitrogen, USA) followed by 4-bromoanisole (BAN Phase Separation Reagent, # BN 191, Molecular Research Center, Inc) and isopropanol (2-Propanol (HPLC), Fisher ChemicalTM, # A451-4, Fisher Scientific, USA) for RNA isolation using published manufacturer’s protocol [36]. cDNA synthesis was carried out from 2 μg of RNA using High-Capacity RNA-to-cDNA kit (# 4387406, Thermo Fisher Scientific, USA). Real-Time qRT-PCR was performed with SYBR Green Mix (PowerUpTM SYBRTM Green Master Mix, # A25741, ThermoFisher Scientific, Waltham, MA, USA), and data were generated by Ct method (2-ΔΔCt). Primers indicated in Table 1.

### 2.6. Cell Isolation and Flow Cytometry

Colonic lamina propria lymphocytes were isolated as described before [37]. Briefly, the colon was removed, cut longitudinally, and washed with Roswell Park Memorial Institute Medium (RPMI), 10% fetal bovine serum (FBS),1 mM HEPES (N-2- hydroxyethylpiperazine-N -2-ethanesulfonic acid; RPMI-10/HEPES). Tissue was further cut into 5 mm × 5 mm sections, followed by 6 washes. Tissues were digested in 2 serial 1-h incubation with collagenase-E (95 IU/mL; Sigma-Aldrich, St Louis, MO, USA), and further disrupted by it passing through a 17-gauge needle and filtered through 40 µm strainer to create single suspensions. Mononuclear cells were enriched by a Percoll gradient. Prior to antibody staining, Fc receptors were blocked by Trustain FcX (Biolegend, San Diego, CA, USA). The following antibodies were used for staining: PeCY7 anti-mouse CD3 (clone-145-2C11, Biolegend, San Diego, CA, USA), FITC anti- mouse CD4 (clone GK 1.5), PacBlue anti-mouse CD8α (Clone-53-6.7, Biolegend), and PE Anti-mouse/Rat FOXP3 (clone FJK-16s, eBioscience). For Foxp3 staining, the eBioscience Foxp3/Transcription set were used as per the manufacturer’s instructions. Samples were analyzed on an Invitrogen “Attune” NxT Flow Cytometer (Thermo Fisher Scientific, Waltham, MA, USA).

### 2.7. Statistical Analysis

The statistical analysis was performed by GraphPad Prism version 8. All data were presented as means ± SEM and the statistically significant differences were considered at *p* < 0.05. The Kruskal–Wallis non-parametric test was used to compare data between all groups.

## 3. Results

The length of the intestine is an important physiological index for intestinal development and health and is affected by nutrient intake [38]. The average SI length for the groups in the current study were as follows; CC = 39.40 cm, DF = 38.03 cm, SW-C = 37.14 cm, CR = 36.09 cm, HFD = 35.86 cm, ADF = 35.28 cm, and TRF = 34.64 cm. Average length for the colon was CC = 6.27 cm, DF = 6.84 cm, SW-C = 6.79 cm, CR = 6.23 cm, HFD = 5.98 cm, ADF = 6.28 cm, and TRF = 5.73 cm. Overall, the groups consuming a high fat diet, irrespective of fasting protocol, had decreased small intestinal and colon length with significant differences between the CC vs. TRF (*p* < 0.01) and vs. ADF (*p* = 0.03) for SI (Figure 2A,B). The intestinal length for the DF group was also significantly longer than TRF (*p* = 0.02) group in SI and the ad libitum HFD (*p* = 0.02) and TRF (*p* < 0.01) in colon (Figure 2A,B).

Consumption of the HFD, irrespective of fasting protocols, resulted in reduced cecum weight as compared to the groups that consumed diets containing a higher fiber content. There were significant differences between the following groups: CC vs. HFD (*p* = 0.01), CR (*p* = 0.01) and ADF (*p* = 0.01), SW-C vs. HFD (*p* < 0.01), CR (*p* < 0.01), ADF (*p* < 0.01) and DF vs. HFD (*p* < 0.01), CR (*p* < 0.01) and ADF (*p* < 0.01) (Figure 2C).

Villi length was examined in the proximal, mid, and distal regions of the SI. The average villi length was consistently shorter in the more distal regions of the SI (average villi length including all animals: proximal region = 433.53 μm, mid region = 347.21 μm, and distal region = 240.19 μm). Minimal differences were observed between groups in the proximal and mid intestinal region (Figure 3A,B). The distal region of the SI was most influenced by dietary protocols: the chow control group had significantly longer villi than HFD (*p* = 0.007), TRF (*p* = 0.03), and ADF (*p* = 0.0001) groups, while the animals that consumed chow after high fat diet exposure (SW-C) had significantly longer villi than the ADF group (*p* = 0.01) (Figure 3C). 

Crypt depth was also measured in both the SI and colon. The crypt depth was consistently decreased with high fat diet consumption with no difference between proximal, mid and distal regions of the SI (Figure 4A–C). The chow control group (CC) which had no high fat diet exposure had the greatest crypt depth of all groups. In the proximal region of the SI the crypt depth of the CC group was significantly different from HFD (*p* = 0.003), DF (*p* = 0.03), TRF (*p* = 0.01), and ADF (*p* = 0.002) groups (Figure 4A). The reduction was also seen in the mid SI region with CC being significantly greater than all high fat consuming groups (*p* < 0.02, Figure 4B) and distal region where CC was significantly greater than the HFD (*p* = 0.001), TRF (*p* = 0.001), and ADF (*p* < 0.0001) groups (Figure 4C). The animals that were switched to the chow diet after the consumption of high-fat diet demonstrated some recovery and had significantly greater crypt depth than HFD (*p* = 0.02), TRF (*p* = 0.02), and ADF (*p* = 0.0002) groups (Figure 4C). The change in crypt depth in the colon mimicked what was seen in the distal SI with HFD, TRF, and ADF significantly decreased compared to chow control group (*p* ≤ 0.01) with some recovery seen in the SW-C group where SW-C was significantly greater than HFD (*p* = 0.004) and TRF (*p* ≤ 0.004) (Figure 4D).

Goblet cells were identified and quantified from microscope images where cells were stained with Periodic Acid-Schiff staining protocol in distal SI and colon only. See Figure 5A insert for representative image of Goblet cell staining. Significant differences were observed in the distal SI villi, where ADF had significantly less goblet cells than the DF (*p* = 0.04) and the TRF group (*p* = 0.02, Figure 5B). This difference in cell number is not due to differences in villi length between these groups as villi length is comparable. Within the crypt region of the distal SI only the SW-C had significantly more cells than the ad libitum HFD group (*p* = 0.03, Figure 5C). The colon of the DF group had significantly more goblet cells than the HFD (*p* = 0.004), CR (*p* = 0.01), and ADF (*p* = 0.001) groups (Figure 5C). 

As the majority of the diet-associated changes was seen in the distal region of the SI, the expression of the immune-associated transcription factors Foxp3 and RORγt were measured at this location. The DF group had the highest level of Foxp3 expression, with significantly more transcript than the CR group (*p* = 0.01, Figure 6A). Although RORγt showed a trend of increased expression in the groups where the high fat diet was consumed, no significant differences were detected (Figure 6B). Consistent with increased Foxp3 expression, the DF diet also resulted in increased Foxp3^+^ cells isolated form the colon in the DF group with significant differences detected between DF and the ADF group (*p* = 0.02, Figure 6C). 

## 4. Discussion

The epithelial surface of intestine is constantly exposed to the external environment that includes dietary components and gut resident microorganisms [39]. The content of the diet and structure of microbiome have been tightly linked to the change in intestinal morphology [22,40]. In this study, we investigated the influence of various diets and feeding strategies on intestinal length, epithelial parameters, and mucosal immunity. The findings in this study are consistent with the previous literature demonstrating that high fat diet consumption alters the microbiome, intestinal morphology, and mucosal immune function [41,42].

The length of SI and colon is crucial for an appropriate digestion and absorption [38]. The results from this study suggested that SI and colon length were greater in high fiber consuming groups vs. high fat consumption. Fasting protocols, when consuming a high fat diet, did not improve these intestinal parameters to the levels seen with high fiber consumption. Cecum weight was also increased when consuming high amounts of fermentable fiber, as this nutrient is able to modulate intestinal transit time, increase intestinal weight, fecal bulk and moisture, and enhancing the bacterial diversity [41,43,44]. In animals, consumption up to 5% of pectin had been shown to increase weight and length of the SI and colon [45,46]. Cecum weight is also significantly increased with the consumption of beet and pea fiber [47], while cellulose has been shown to increase colon length [46,48]. Inulin, and high amylose maize starch are also able to increase the length of large intestine significantly in comparison to a highly digestible starch diet [49,50]. The Daniel Fast diet in the current study is rich in cellulose, inulin, and corn starch Hi-Maize 260 [2,34] and consumption of this diet modulate gut microbiota specifically improving bacterial diversity [43], and increasing abundance of genera such as *Akkermansia Muciniphila* which has gut protective functions through maintenance of intestinal integrity [51,52]. All HFD groups had increased levels of Firmicutes compared to the DF group, which were enriched in Bacteroidetes [34]. The increase in ratio of Firmicutes to Bacteroidetes had been linked to inflammatory bowel diseases, such as (ulcerative colitis and Crohn’s disease), irritable bowel syndrome, and intestinal epithelium disintegration [22,53,54]. 

Although subtle differences were observed in villi length in the proximal and mid SI between groups, the distal SI was most significantly influenced by diet. Consumption of the high fat diet reduced villi length, with only partial recovery of villi when changing to the chow, not seen with any of the other dietary protocols. In animal studies, high fat diets have been shown to result in shortened villi with a loss in epithelial integrity in comparison to chow control diet [22,55]. 

The crypt mainly functions as an architectural unit of the stem cell niche. Its structure protects stem cells from luminal content and provides the required number of amplifying cells [56]. Crypts are also dynamic structures that change postnatally and undergo multiple rounds of replication via a process of crypt fission [57]. High fat diet consumption results in colonic crypt depth reduction [23,58], and in the current study, crypt depth was decreased in high fat consuming animals, with some recovery seen when switched back to chow diet. Interestingly, the recovery was not as robust when switching to the DF diet. The recovery in crypt depth seen in this study when the HFD was replaced with the chow (SW-C) most likely resulted from the reduction of fat in the diet in combination with the increased fiber. Increased consumption of dietary pectin, fructooligosaccharide, and inulin has been shown to increase crypt depth in SI and colon [59,60,61].

Goblet cell numbers were also measured in the current study. Groups consuming the HFD had reduced goblet cells, while consumption of high fiber diets after high fat exposure (DF and SW-C) increased the number of goblet cells in the villi and crypts. This is consistent with previous studies showing that dietary fiber increases the number of mucin secreting goblet cells along the intestine [62]; feeding rats different types of soluble and insoluble fibers, including konjac mannan (95%), guar gum (81.5%), psyllium (90%), wheat bran (77%), and beet fiber (78%), increased the number of PAS-stained goblet cells [63]. Carboxymethylcellulose (CMC) [64], potato fiber and potato-resistant starch diets [65], and fermentable oligosaccharides from either broccoli fiber (*Brassica oleracea L. var. italica*) or microcrystalline cellulose also increased the number of goblet cells [66]. It is important to mention that the mucin-degrading bacterial strain *Akkermansia Muciniphila* is increased when animals are fed the DF diet [34]. The increase in this genus in response to the consumption of dietary fiber was reported in several studies [67,68]. Colonization of *A. muciniphila* stimulates mucin production from goblet cells [69]. Interestingly, administration of *A. muciniphila* accelerate the proliferation of Lgr5^+^ of intestinal stem cells and increase the differentiation of goblet cells in SI and colon [70]. 

Intestinal-associated immune cells can also be impacted by the diet content and timing, and the gut microbiome [71,72]. HFD for example was associated with the increase of intestinal proinflammatory cytokines IL-1β, IL-6, IFNγ, and TNFα [73,74]. In contrast, a low-fat/high-fiber diet alleviated the inflammatory influence of HFD consumption [75]. Administration of 4% soybean fiber into the total intake was enough to down-regulate the expression of proinflammatory cytokines IL-8, IL-1β, and TNFα, while administration up to 8% of soybean fiber up-regulates the expression of anti-inflammatory cytokines, such as IL-10 and TGF-β1 [76]. Results from the current study suggest that the DF diet increased the number Foxp3^+^Tregs cells in the SI and colon. Foxp3 is essential for the cellular differentiation of regulatory T cells (Tregs) [77] and is also able to regulate differentiation of Th17 cells by binding into the RORγt protein and antagonizing its binding to DNA [78]. Although not statistically significant, the expression level of RORγt was consistently higher in all HFD group. These findings suggest that the content of DF diet (most likely the fiber content and high amount of omega-3 fatty acids) is able to modulate the mucosal immune system. At this point it is not clear if it acts directly on the immune cells or if changes are induced through microbiome alterations. Previous work had shown dietary fiber increases intestinal Foxp3 expression and Tregs cells [79,80]. Feeding long chain inulin-type fructans to NOD/LtJ mice elevated colonic Tregs cells and levels of IL-10 production [80]. Short-chain fatty acids (SCFAs), bacterial fermentation products of dietary fibers, can also modulate and increase the regulatory T cells number through signaling into their metabolite-sensing G-protein coupled receptors (GPCRs) [81]. Administration of SCFAs in the drinking water of germ-free mice increased Tregs frequency and number in the colon that mediated through GPCR43, which is expressed on colonic Tregs [82]. Consistent with these finding, the DF group had high concentration of SCFAs [34].

## 5. Conclusions

In the current study we investigated the influence of various diets and fasting protocols on intestinal-associated parameters that is altered by the consumption of a high fat diet. Although all fasting protocols used in this study have previously been shown to improve metabolic outcomes such as excess weight and glucose intolerance, these protocols did not result in improvement in gut parameters to the extent that is seen with diets containing higher amounts of fiber. We conclude that fasting protocols, while consuming a high fat diet, do not result in improvement in the intestinal health and immune parameters.

## Figures and Tables

**Figure 1 medicines-10-00018-f001:**
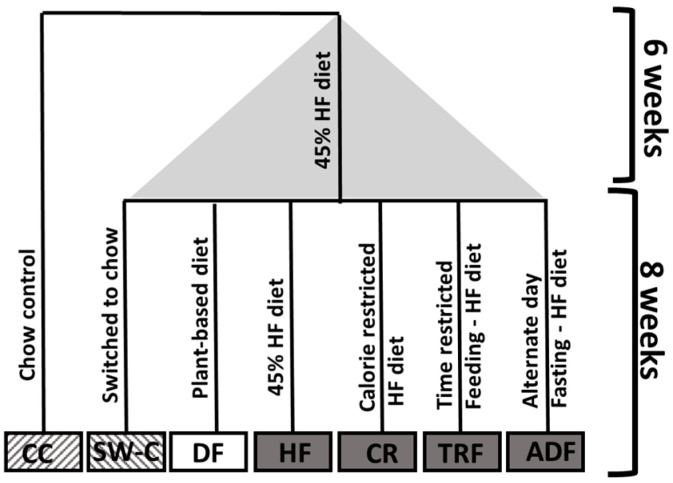
Experimental group set up.

**Figure 2 medicines-10-00018-f002:**
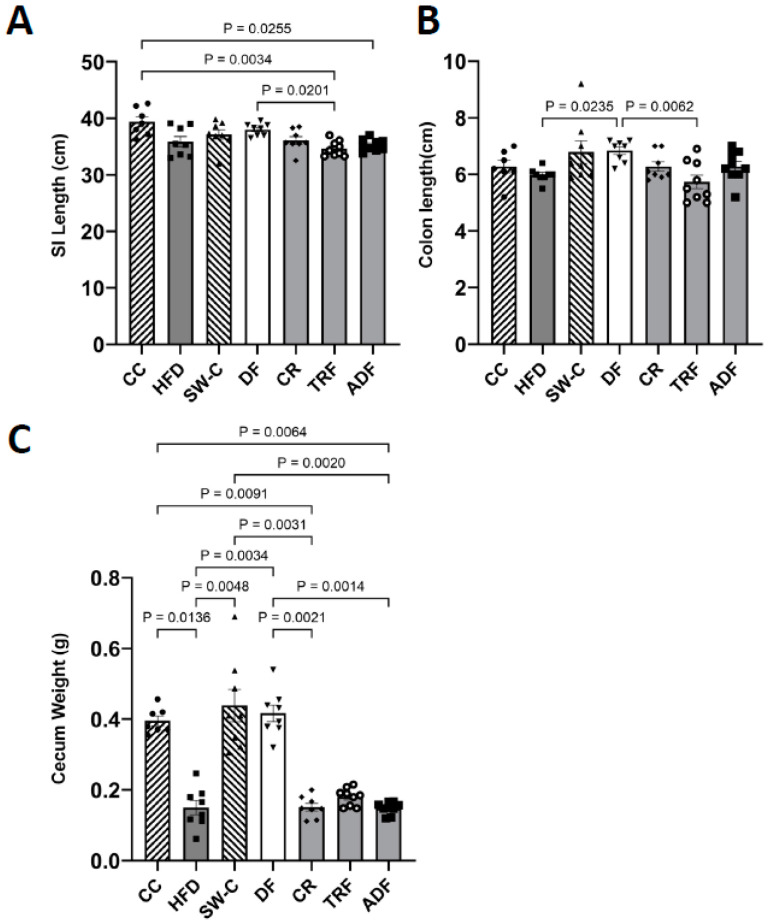
Differences in length of small intestine (**A**) and colon length (**B**), and cecum weight (**C**) induced by dietary protocols. Values are mean ± SEM, *p* < 0.05 indicated on graph. Significant differences are determined by Kruskal–Wallis test. CC, Chow Control (*n* = 7); HFD (*n* = 8), High Fat Diet; SW-C, Switch to Chow (*n* = 8); DF, Daniel Fast (*n* = 8); CR, Caloric-Restricted (*n* = 8); TRF, Time-Restricted Feeding (*n* = 9); and ADF, Alternate-Day Fasting (*n* = 9).

**Figure 3 medicines-10-00018-f003:**
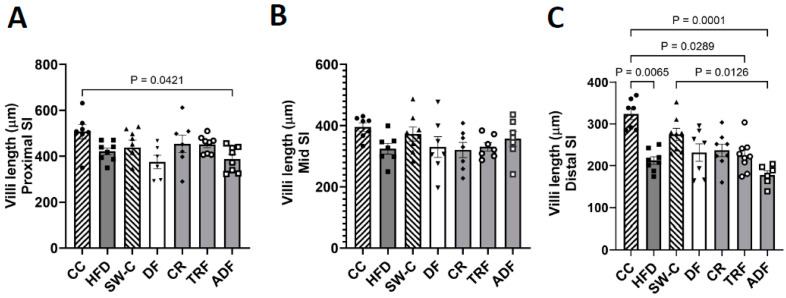
Differences induced in villi length within the proximal (**A**), mid (**B**) and distal (**C**) small intestine in response to dietary protocols. Values are mean ± SEM, *n* = 6–9 per group. *p* < 0.05 indicated on graph. Significant differences are determined by Kruskal–Wallis test. CC, Chow Control; HFD, High Fat Diet; SW-C, Switch to Chow; DF, Daniel Fast; CR, caloric-restricted; TRF, time-restricted feeding; and ADF, Alternate-Day Fasting.

**Figure 4 medicines-10-00018-f004:**
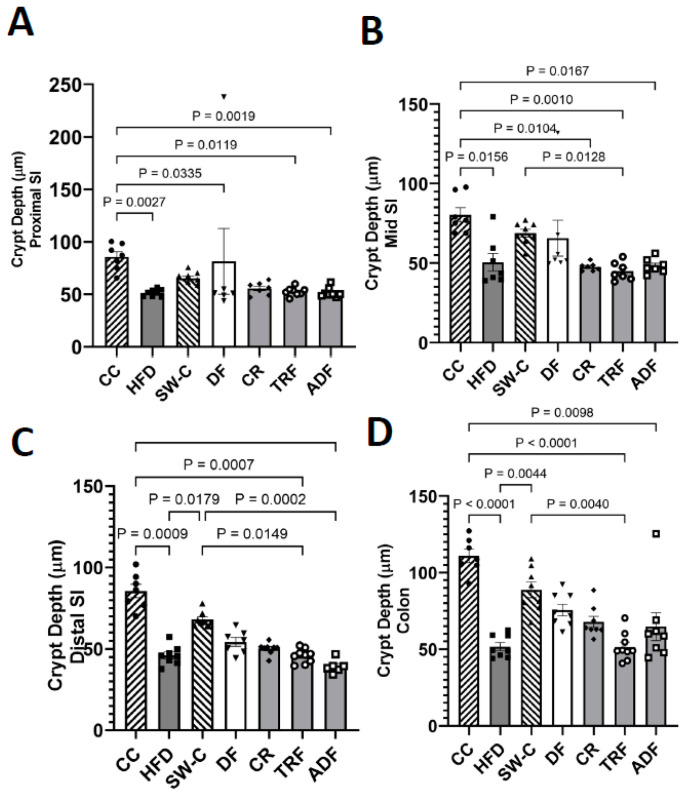
Dietary protocol induced differences in crypt depth in proximal (**A**), mid (**B**) and distal (**C**) small intestine and colon (**D**). Values are mean ± SEM, *n* = 6–9 per group. *p* < 0.05 indicated on graph. Significant differences are determined by Kruskal–Wallis test. CC, Chow Control; HFD, High Fat Diet; SW-C, Switch to Chow; DF, Daniel Fast; CR, Caloric-Restricted; TRF, Time-Restricted Feeding; and ADF, Alternate Day Fasting.

**Figure 5 medicines-10-00018-f005:**
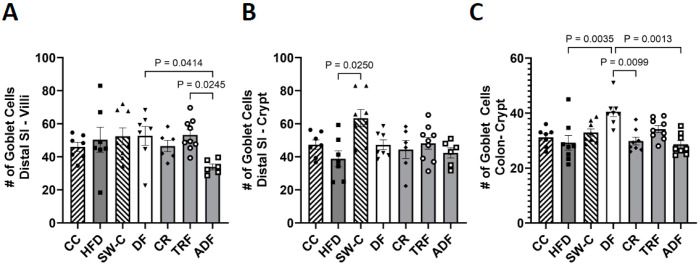
Dietary protocol induced differences in Goblet cell numbers in the distal SI (**A**,**B**) and colon (**C**). Values are mean ± SEM, *n* = 6–9 per group. *p* < 0.05 indicated on graph. Significant differences are determined by Kruskal–Wallis test. CC, Chow Control; HFD, High Fat Diet; SW-C, Switch to Chow; DF, Daniel Fast; CR, Caloric-Restricted; TRF, Time-Restricted Feeding; and ADF, Alternate-Day Fasting.

**Figure 6 medicines-10-00018-f006:**
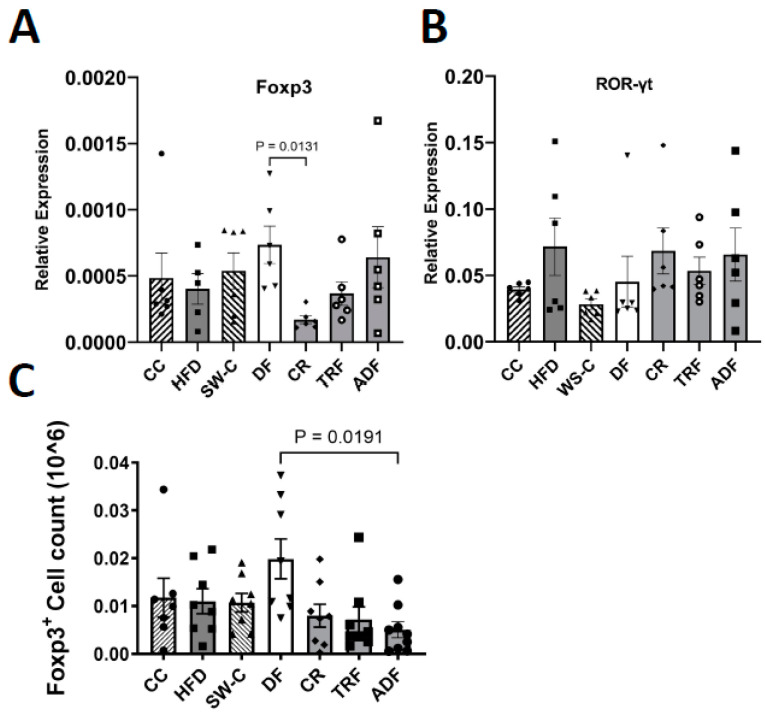
Relative expression of Foxp3 (**A**) and RORγt (**B**) in distal SI. (**C**) Absolute cell count of CD4^+^Foxp3^+^ cells isolated from the colon. Values are mean ± SEM, *n* = 6–9 per group. *p* < 0.05 indicated on graph. Significant differences are determined by Kruskal–Wallis test. CC, Chow Control; HFD, High Fat Diet; SW-C, Switch to Chow; DF, Daniel Fast; CR, Caloric-Restricted; TRF, Time-Restricted Feeding; and ADF, Alternate-Day Fasting.

**Table 1 medicines-10-00018-t001:** Primers.

Primer Name	Primer Sequence (5′–3′)
β-actin	Fwd: ACCTTCTACAATGAGCTGCG Rev: CTGGATGGCTACGTACATGG
ROR-γt	Fwd: AAGTACCACAATATGCGACCC Rev: TCTGAAGTAGGCGAACATGC
Foxp3+	Fwd: TTTCTGAGGATGAGATTGCCC Rev: TTGTCGATGAGTCTTGCAGAG

## Data Availability

The data presented in this study are available on request from the corresponding author.

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
