# Peer review of "Fasting Protocols Do Not Improve Intestinal Architecture and Immune Parameters in C57BL/6 Male Mice Fed a High Fat Diet"

_medicines, 2023, doi:10.3390/medicines10020018_

Round 1

Reviewer 1 Report

I read the manuscript by Raed et al., titled “Fasting Protocols Do Not Improve Intestinal Architecture and Immune Parameters in C57BL/6 Male Mice Fed a High Fat Diet”

This is an experimental study conducted with the intention of evaluating various fasting protocols after ad libitum high-fat diet (HFD) consumption on intestinal morphology and mucosal immunity. Authors found that consuming a HFD resulted in the shortening of the intestine and a reduction in villi and crypt size. Fasting, while consuming the HFD, did not restore these parameters to the extent seen with the chow and DF diet. Goblet cell number and gut-associated regulatory T cells had improved recovery with the high fiber diets, not seen with the HFD, irrespective of fasting.

The study was interesting to read, and it has the potential to contribute to the research community and the field. Although it is quite an interesting field and direction of research, I have noted some concern that needs the authors’ attention to improve the quality of the manuscript.

Some of the concerns are highlighted below:

Abstract

1.       Page 1, line 8-9: The phrase “fasting protocol” might be misleading to readers as these animals were randomized to receive a diet for example group 1 and 2 there was no fasting indicated.

2.       Randomization to different dietary protocols was it for 2 months? If so, specify.

3.       Page 1 line 15: “Small intestine”, are the researchers referring to the entire small intestine or they measured the duodenum, jejunum or ileum?

Introduction

1.       It is well written and supported by evidence as supporting references which are updated as new and old evidence.

2.       Some of the minor comments here include typographical or syntax errors and abbreviations without being written in full. For example

3.       Page 2, line 47: LDL write in full for the first-time use and abbreviate subsequently.

4.       Page 2, line 51: correct use of word ‘‘effect”

5.       Page 2, line 73: FOX p3 and ROR-yt

6.       Page 2 lines 76, 81: IL and TNF-α.

7.       What is  IL-17F ?

Methods and material.

1.       Despite the results of cecum weight been presented, its not shown how cecum weight was measured. I suggest authors indicate briefly how they measured the weight of the cecum (Figure 1 C).

2.       Page 2, Line 95: Ethics number should be provided,  and the reporting should follow ARRIVE guidelines since animals were used.

3.       Page 3, lines 104: 8-9 is not specific as to exactly what the researcher has used.  So, I suggest they indicate exactly the number they used in their study (recent study). Given that here they have 6 groups, how was the number of animals divided per group? This should be clear and specified.

4.       Page 3, line 113: Again, this must be specific to their study. Was it 8 or 9 weeks? If one group was 8 or 9 this should be specified as to which group was fed for 8 or 9 weeks this is transparency and reproducibility by other researchers.

5.       Page 3, line 115: Avoid non-standardized abbreviations. For example, use of SI, for example, you cannot write an abbreviation first and then write it in full later. It would be better if authors want to abbreviate this to write it in full for first time and abbreviate subsequently and not visa versa.

6.       Page 3, line 144: “11 to 18 measurements….”. The sentence may start with a number if it is written in a word. Revise by writing numbers in words.

7.       Page 3, line 145: Small intestine is already written in full and abbreviated subsequently.

8.       Page 4, line 164: RPMI, FBS write in full.

Statistical analysis

1.       Page 4, line 177: Does this means all analyzed data were nonparametric? I would have preferred non-parametric data to be reported as median and interquartile range. This is partly because the mean is not always an ideal measure of the central tendency of a sample.

Results

1.       Page 5, Line 182: Already abbreviated no need to write in full here to avoid redundancy abbreviate here.

2.       Page 5 line 183: Authors have neatly and visibly presented the data for small intestinal length for all groups on page 5, lines 183-184, and this is also presented by figure  A. I suggest this be uniform for colon length as results are only shown in Figure B which to others might not be informative rather than showing significant difference. Giving the value will show the magnitude of the differences among the groups.

3.       Was histological analysis, and images of the small intestine taken during analysis? This could be more informative even if provided as supplementary material if authors are worried about the length of the manuscript. Although authors indicate that “The EVOSTM M7000 Imaging System was used for imaging and Invitrogen Celeste Image Analysis Software (SKU# AMEP4816) used for measurements of the morphological structures”. The exact histological, imaging or morphological results are not presented.

4.       Page 6, line 186: For consistency in reporting, this p-value should be 0.00; otherwise, write as it is from the software and be consistent in terms of rounding off throughout the manuscript.

5.       Page 5 line 188: For consistency, this should be reported as 0.01 otherwise write in full as computed by software but remember consistency throughout the manuscript.

6.       Taking a closer look at Figure 1 B, the length of the colon, it seems the comparison was made between the HFD group and DF  and DF vs TRF. What was the purpose of having a control group in this case? Isn’t comparison supposed to be within and across all groups to make a concrete conclusion? This also applies to Figure 1A.

7.       Figure 2 legend in terms of animals per group needs to be revised as it is confusing. The 9-6 animals per group don’t make sense. Maybe authors need to specify the exact number used per group in this study rather than generalization.

8.       Table 1 presented in this manuscript shows different primers; it would have been great to also determine the expression of these inflammatory cytokines, including IL-10, 6, and TNF-α in addition to FoXp3+.

Discussion

1.       Page 8, lines 279-280: “The results from this study showed that SI and colon length were increased in high fiber consuming groups, while fasting protocols when consuming a high-fat diet, did not alter these parameters.” This statement contradicts the finding reported in this study, as shown in figure 1. Figure 1 indicates that all dietary groups had decreased small intestinal length. This need to be revised or clarified more.

Conclusion

1.       This conclusion is not informative; there is no link between the objective and the conclusion drawn from the study; the major focus should be on intestinal health and immunity/immune parameters. This need revision.

Author Response

We thank the reviewer for the careful review of the current manuscript. We appreciate the effort that goes into a thorough manuscript review. 

Below we have addressed all comments by reviewer 1. Our replies are indicated in bold. 

Abstract

  1. Page 1, line 8-9: The phrase “fasting protocol” might be misleading to readers as these animals were randomized to receive a diet for example group 1 and 2 there was no fasting indicated.

“Fasting protocol” was changed to “dietary protocols” to reflect that it was not only fasting protocols that was used in the study.

  1. Randomization to different dietary protocols was it for 2 months? If so, specify.

Animals were randomized to various protocols for two months. This is now clarified in the abstract as well as the method section.   

  1. Page 1 line 15: “Small intestine”, are the researchers referring to the entire small intestine or they measured the duodenum, jejunum or ileum?

In this instance we are referring to the entire intestine as both the small intestine and colon showed reduction in length when consuming a high fat diet. 

Introduction

  1. It is well written and supported by evidence as supporting references which are updated as new and old evidence.

Thank you for this comment.

  1. Some of the minor comments here include typographical or syntax errors and abbreviations without being written in full. For example
  2. Page 2, line 47: LDL write in full for the first-time use and abbreviate subsequently.

We wrote out the full name for LDL – low-density lipoprotein.

  1. Page 2, line 51: correct use of word ‘‘effect”

“Effect” is changed to “affect”.

  1. Page 2, line 73: FOX p3 and ROR-yt

Abbreviations are defined.

  1. Page 2 lines 76, 81: IL and TNF-α.

Abbreviations are defined.

  1. What is  IL-17F ?

 Please see “IL-17F: regulation, signaling and function in inflammation” by Chang and Dong. (Cytokine, 2009)

Methods and material.

  1. Despite the results of cecum weight been presented, its not shown how cecum weight was measured. I suggest authors indicate briefly how they measured the weight of the cecum (Figure 1 C).

We added a description of cecum weight determination in the method section (lines 119-121)

  1. Page 2, Line 95: Ethics number should be provided,  and the reporting should follow ARRIVE guidelines since animals were used.

The following was added to the methods: “All experiments were approved by the University  of Memphis Institutional Animal Care and Use Committee (Protocol #0806).”

  1. Page 3, lines 104: 8-9 is not specific as to exactly what the researcher has used.  So, I suggest they indicate exactly the number they used in their study (recent study). Given that here they have 6 groups, how was the number of animals divided per group? This should be clear and specified.

The number of animals in each group is now defined with group description in the methods section. 

  1. Page 3, line 113: Again, this must be specific to their study. Was it 8 or 9 weeks? If one group was 8 or 9 this should be specified as to which group was fed for 8 or 9 weeks this is transparency and reproducibility by other researchers.

In the manuscript we now state that after 8 weeks, the animals were sacrificed over a 3-day period (line 117-118) The reason for the multiple day sacrifice was to ensure that all tissues were collected within a small window of time to not have circadian rhythm changes as an additional confounding factor. Animals were sacrificed such that all groups were represented each day during the harvest.  

  1. Page 3, line 115: Avoid non-standardized abbreviations. For example, use of SI, for example, you cannot write an abbreviation first and then write it in full later. It would be better if authors want to abbreviate this to write it in full for first time and abbreviate subsequently and not visa versa.

Thank you for this observation. SI is now corrected throughout the document.

  1. Page 3, line 144: “11 to 18 measurements….”. The sentence may start with a number if it is written in a word. Revise by writing numbers in words.

The numbers are now written in words.

  1. Page 3, line 145: Small intestine is already written in full and abbreviated subsequently.

Thank you for pointing this out. It is now corrected throughout the manuscript.

  1. Page 4, line 164: RPMI, FBS write in full.

RPMI and FBS are now written in full.

Statistical analysis

  1. Page 4, line 177: Does this means all analyzed data were nonparametric? I would have preferred non-parametric data to be reported as median and interquartile range. This is partly because the mean is not always an ideal measure of the central tendency of a sample.

Due to the limited sample sizes, we used a non-parametric test to determine significance.

Results

  1. Page 5, Line 182: Already abbreviated no need to write in full here to avoid redundancy abbreviate here.

SI is now used throughout the manuscript.

  1. Page 5 line 183: Authors have neatly and visibly presented the data for small intestinal length for all groups on page 5, lines 183-184, and this is also presented by figure  A. I suggest this be uniform for colon length as results are only shown in Figure B which to others might not be informative rather than showing significant difference. Giving the value will show the magnitude of the differences among the groups.

 Average colon length is now also added in the text to be consistent with the SI data presentation (line 194-196).

  1. Was histological analysis, and images of the small intestine taken during analysis? This could be more informative even if provided as supplementary material if authors are worried about the length of the manuscript. Although authors indicate that “The EVOSTM M7000 Imaging System was used for imaging and Invitrogen Celeste Image Analysis Software (SKU# AMEP4816) used for measurements of the morphological structures”. The exact histological, imaging or morphological results are not presented.

The exact method for the histological analysis is now added to the method section (line 156-165). We chose not to  add images to the manuscript as the amount of variables (proximal, mid, and distal small intestine and colon) for all seven groups made the manuscript too long. The quantitation on the histological results were more informative. 

  1. Page 6, line 186: For consistency in reporting, this p-value should be 0.00; otherwise, write as it is from the software and be consistent in terms of rounding off throughout the manuscript.

All p-values less than 0.01 are now indicated as p<0.01.

  1. Page 5 line 188: For consistency, this should be reported as 0.01 otherwise write in full as computed by software but remember consistency throughout the manuscript.

All p-values less than 0.01 are now indicated as p<0.01.

  1. Taking a closer look at Figure 1 B, the length of the colon, it seems the comparison was made between the HFD group and DF  and DF vs TRF. What was the purpose of having a control group in this case? Isn’t comparison supposed to be within and across all groups to make a concrete conclusion? This also applies to Figure 1A.

All groups were compared to each other using Dunn’s multiple comparison test to determine what effect the diet and fasting has on the intestinal parameters. The CC (chow control) group was included to demonstrate what these parameters look like is a healthy age-and sex matched mouse population.

  1. Figure 2 legend in terms of animals per group needs to be revised as it is confusing. The 9-6 animals per group don’t make sense. Maybe authors need to specify the exact number used per group in this study rather than generalization.

Sample size for each group is now defined in methods. As sample size is also indicated by the data points on the graph, we used the sample size range to demonstrate sample size for the specific datasets in figure legends.

  1. Table 1 presented in this manuscript shows different primers; it would have been great to also determine the expression of these inflammatory cytokines, including IL-10, 6, and TNF-α in addition to FoXp3+.

IL-10, IL-6 and TNF- α was not measured in this study and the primer sequences was removed. We agree that it would have been interesting to measure these cytokines. 

Discussion

  1. Page 8, lines 279-280: “The results from this study showed that SI and colon length were increased in high fiber consuming groups, while fasting protocols when consuming a high-fat diet, did not alter these parameters.” This statement contradicts the finding reported in this study, as shown in figure 1. Figure 1 indicates that all dietary groups had decreased small intestinal length. This need to be revised or clarified more.

This sentence is now clarified and read as follows:” The results from this study suggested that SI and colon length were greater in high fiber consuming groups vs high fat consumption.   Fasting protocols, when consuming a high fat diet, did not improve these intestinal parameters to the levels seen with high fiber consumption.”

Conclusion

  1. This conclusion is not informative; there is no link between the objective and the conclusion drawn from the study; the major focus should be on intestinal health and immunity/immune parameters. This need revision.

Thank you for pointing out the disconnect between the objective and conclusion. 

The conclusion is revised to now state the following: “In the current study we investigated the influence of various diets and fasting protocols on intestinal-associated parameters that is altered by the consumption of a high fat diet. Although all fasting protocols used in this study have previously been shown to improve metabolic outcomes such as excess weight and glucose intolerance, these protocols did not result in improvement in gut parameters to the extent that is seen with diets containing higher amounts of fiber. We conclude that fasting protocols, while consuming a high fat diet, do not result in improvement in the intestinal health and immune parameters.

Reviewer 2 Report

The authors aimed to evaluate various fasting protocols after ad libitum high-fat diet consumption on intestinal morphology and mucosal immunity. Authors have found that diets containing higher amounts of fiber and omega-3 fatty acids improved gut parameters; however, fasting protocols failed to improve gut parameters.

Comments

The authors described the experimental protocol and groups well in the materials and methods section. Still, it would greatly help if they included a protocol figure in the manuscript.

Did the authors examine the effect of different feeding protocols on body mass, fat mass, and lean mass?

Did the authors examine the effect of different feeding protocols on glucose-related or lipid parameters?

Did the authors examine the hormone-releasing enteroendocrine cells?

Did the authors use HE-stained slides for villus height and crypt depth measurements?

Author Response

We thank the reviewer for the comments. Below all comments are addressed in bold. 

The authors described the experimental protocol and groups well in the materials and methods section. Still, it would greatly help if they included a protocol figure in the manuscript.

A figure of the experimental set up is now added as Figure 1. 

Did the authors examine the effect of different feeding protocols on body mass, fat mass, and lean mass?

Yes, and this data has been published. See Smith et. al. A comparison of dietary and caloric restriction models on body composition, physical performance, and metabolic health in young mice: Nutrients 2019, 11(2), 350

Did the authors examine the effect of different feeding protocols on glucose-related or lipid parameters?

Yes, and this data has been published. See Smith et. al. A comparison of dietary and caloric restriction models on body composition, physical performance, and metabolic health in young mice: Nutrients 2019, 11(2), 350

Did the authors examine the hormone-releasing enteroendocrine cells?

Hormone-releasing enteroendocrine cells were not assessed in this particular study.

Did the authors use HE-stained slides for villus height and crypt depth measurements?

Yes, the following was added to the method section: “Hematoxylin and eosin-stained slides were used for villi length and crypt depth determination.” 

Reviewer 3 Report

The investigation is interesting, the authors evaluated the effect HFD-different dietary protocols on intestinal morphology and mucosal immunity.

I consider that include representative images about intestinal morphology changes could be appropriate and important for this manuscript.

Some references included in introduction were published before 2017, this could be updated to more current references.

Author Response

We thank the reviewer for the insightful comments. 

Below we have addressed the comments in bold. . 

I consider that include representative images about intestinal morphology changes could be appropriate and important for this manuscript.

We considered adding images, but as we looked at many variables (proximal, mid, distal small intestine and colon for vili and crypt and goblet cells) and included 7 groups, the manuscript becomes too long when including representative figures.  

Some references included in introduction were published before 2017, this could be updated to more current references.

References are now updated to include more recent publications.